# Towards New Diagnostic Approaches in Disorders of Consciousness: A Proof of Concept Study on the Promising Use of Imagery Visuomotor Task

**DOI:** 10.3390/brainsci10100746

**Published:** 2020-10-17

**Authors:** Antonino Naro, Rocco Salvatore Calabrò

**Affiliations:** IRCCS Centro Neurolesi Bonino Pulejo, SS113 C.da Casazza, 98124 Messina, Italy; g.naro11@alice.it

**Keywords:** chronic disorders of consciousness (DoC), electroencephalography (EEG), minimally conscious state (MCS), motor imagery (MI), unresponsive wakefulness syndrome (UWS), visuomotor integration (VMI)

## Abstract

Background: advanced paraclinical approaches using functional neuroimaging and electroencephalography (EEG) allow identifying patients who are covertly aware despite being diagnosed as unresponsive wakefulness syndrome (UWS). Bedside detection of covert awareness employing motor imagery tasks (MI), which is a universally accepted clinical indicator of awareness in the absence of overt behavior, may miss some of these patients, as they could still have a certain level of awareness. We aimed at assessing covert awareness in patients with UWS using a visuomotor-guided motor imagery task (VMI) during EEG recording. Methods: nine patients in a minimally conscious state (MCS), 11 patients in a UWS, and 15 healthy individuals (control group—CG) were provided with an VMI (imagine dancing while watching a group dance video to command), a simple-MI (imagine squeezing their right hand to command), and an advanced-MI (imagine dancing without watching a group dance video to command) to detect command-following. We analyzed the command-specific EEG responses (event-related synchronization/desynchronization—ERS/ERD) of each patient, assessing whether these responses were appropriate, consistent, and statistically similar to those elicited in the CG, as reliable markers of motor imagery. Results: All patients in MCS, all healthy individuals and one patient in UWS repeatedly and reliably generated appropriate EEG responses to distinct commands of motor imagery with a classification accuracy of 60–80%. Conclusions: VMI outperformed significantly MI tasks. Therefore, patients in UWS may be still misdiagnosed despite a rigorous clinical assessment and an appropriate MI assessment. It is thus possible to suggest that motor imagery tasks should be delivered to patients with chronic disorders of consciousness in visuomotor-aided modality (also in the rehabilitation setting) to greatly entrain patient’s participation. In this regard, the EEG approach we described has the clear advantage of being cheap, portable, widely available, and objective. It may be thus considered as, at least, a screening tool to identify the patients who deserve further, advanced paraclinical approaches.

## 1. Introduction

The JFK Coma Recovery Scale-Revised (CRS-R) is considered as the gold standard in the assessment of behavioral responsiveness of patients with chronic disorder of consciousness (DOC), including the differential diagnosis of unresponsive wakefulness syndrome (UWS) and the minimally conscious state (MCS) [1,2,3,4,5]. Consistently, the awake patients who show a complete behavioral unawareness of the self and the surrounding environment are labeled as in UWS, whereas those with fluctuating, inconsistent, but discernible behavioral responsiveness are labeled as in MCS [2,3,5]. Furthermore, patients in MCS—only demonstrate non-reflex behaviors, whereas patients in MCS+ show command following [4].

The different degree of behavioral responsiveness of patients with DoC is hypothesized to mainly depend on the severity of cortical-thalamo-cortical connectivity breakdown, which affects the capacity of the brain to operationally switch from the external awareness network (EAN) and the default-mode network (DMN) [6,7,8,9,10,11]. However, a non-negligible percentage of patients in DoC is still misdiagnosed, i.e., some patients apparently in UWS are instead in MCS. In other words, some patients may be aware despite their inability to manifest it behaviorally [4,12,13,14,15,16,17,18,19,20,21]. Specifically, these patients complain of a clinical–paraclinical dissociation of awareness, which has been variously referred to covert awareness [19,20], cognitive-motor dissociations [21], functional locked-in syndrome [4,13,14,15,22], MCS* [23,24], non-behavioral MCS [25], or higher-order cortex motor dissociation [26]. This misdiagnosis rate may depend on the fact that the clinical assessment may not be sufficient to detect only subtle differences in behavioral responsiveness, which could be instead assessed by using paraclinical approaches, including EEG and functional neuroimaging (e.g., fMRI). In particular, previous works focused on DoC differential diagnosis and the identification of covertly aware patients by looking at specific EEG (including event-related potentials, connectivity measures, and transcranial magnetic stimulation effects on high-density EEG) and functional MRI features induced by either passive (resting state), stimulation (e.g., pictures, names, transcranial electric or magnetic stimuli), or active paradigms (command following) [12,19,25,27,28,29,30,31,32,33,34,35,36,37,38,39]. Overall, whether a behaviorally unresponsive patient shows positive responses (e.g., EEG power changes, event related spectral perturbations) to an active task (e.g., playing tennis or navigation) and/or demonstrate the functional integrity of awareness-related brain network (by using passive and stimulation-based paradigms), he/she could be covertly aware [4,12,13,14,15,16,17,18,19,20,21,40]. Even though these approaches can encounter a number of methodological (e.g., lack of portability and availability, method contraindication, and analysis procedure biases) and patient-related limitations (e.g., arousal fluctuations, varied causes of brain injuries, cognitive impairment, and attentional and sensorimotor deficits) accounting for both false-negative and false-positive cases [41,42], these are proposed as valid methods for detecting covert awareness in patients with DoC, particularly by investigating any residual ability to command-following in active paradigms of mental imagery [23]. Actually, the task involving motor imagery allows detecting command-following, a universally accepted clinical indicator of awareness in the absence of overt behavior [12]. However, a negative response to an active paradigm does not necessarily exclude awareness, as the individual could still have a certain level of self-awareness. Therefore, to identify other new objective and reliable biomarkers of covert awareness is mandatory.

Visual pursuit (or visual tracking) and visual fixation are frequently the first signs of awareness recovery [30,43]. Both are evaluated using the CRS-R. However, both visual fixation and visual pursuit can fail to diagnose patients’ behavioral awareness [30,43]. Therefore, whether visual pursuit and visual fixation are signs of consciousness remains still debatable [42]. Notwithstanding, detecting awareness in visuomotor as well as other sensorimotor processes is significantly important concerning prognosis (as it would be more favorable), patients’ management (as they could also experience pain, could benefit more from early, intensive neurorehabilitation, deep brain stimulation, and pharmacologic therapies), and end-of-life decisions [44]. Therefore, several eye movement measurement-related approaches have been conducted to improve the diagnostic assessment in DoC [45]. Particularly, delta, alpha, and beta blink-related oscillations (BROs) [46], and the assessment of visuomotor integration [47,48] provided reliable information concerning DoC differential diagnosis. However, both these approaches do not reliably tell us whether visual awareness is preserved. The BROs suggest that patients with MCS but not those with UWS have a bottom-up driven activation of the EAN and thus are prone to behavioral responsiveness but in an almost unintentional way. In contrast, visuomotor integration indicates that potentiation of the EEG signs of visuomotor and premotor–motor connectivity may be appreciable in patients with MCS but not in those with UWS.

Notably, visual awareness relies on the functional preservation of the connectivity within a large-scale visuomotor frontoparietal-occipital network (FPON) [49]. Therefore, demonstrating the reliably and repeatedly capacity to modulate functional EEG responses related to visuomotor FPON using an active visuomotor task could reflect preserved awareness. We tried to establish a proof of concept: it is possible to demonstrate differential command following (as reflected by the reliable and repeated capacity of modulating functional EEG responses) as tested within a single recording session to confirm (or even improve) awareness detection in DoC patient. Therefore, our study aimed to assess the individual ability to generate willful EEG responses to a visuomotor imagery task (VMI) by studying task-induced EEG oscillation power changes [12], which has never been tested, to the best of our knowledge [50]. We intentionally used a simple EEG recording device (21 channels) to make the approach cheaper, more accessible, and fully comfortable in comparison with the currently available advanced assessment techniques, considering this experimental paradigm as, at least, a screening tool to identify the patients who deserve further, advanced paraclinical approaches (neurophysiologic and functional neuroimaging). Such a demonstration is important concerning DoC differential diagnosis. Residual structural preservation of visuomotor FPON is the prerequisite for awareness to recover potentially [47,48]. Nonetheless, the functional preservation of the visuomotor FPON demonstrated by a willful EEG response modulation is mandatory for awareness to emerge/recover, thus allowing differentiating patients with MCS (who could show such functional preservation) from those with UWS (who could not show such functional preservation).

## 2. Materials and Methods

### 2.1. Participants

Twenty-eight patients were attending a long-term rehabilitation unit of our Institute, who met the international criteria for UWS and MCS diagnosis [1,2,3,4,5] (as inclusion criterion), were screened for study eligibility. The awareness level was assessed for one month using the CRS-R, which was administered every three days by two assessors expert in CRS-R administration [51,52,53,54,55]. The exclusion criteria were as follows: a DOC condition lasting less than three months after the brain injury [47,48,52,56,57,58,59]; pre-existing severe neurological or systemic diseases; actual critical conditions (i.e., inability to breathe independently, hemodynamic instability); intake of cortical excitability-modifying drugs, beyond L-DOPA, baclofen, and antiepileptic drugs; absence of conventional visual (using a flash lamp at 3 Hz) and auditory evoked potential (using click stimuli delivered at 11 Hz). 

The enrolled sample consisted of 11 UWS and 9 MCS patients consistently with their CRS-R scores. Clinical-demographic characteristics are reported in Table 1. In addition, we included 15 age- and gender-matched healthy control group (CG) (mean age 45 ± 9 years, 9 females and 6 males). 

The Local Ethics Committee approved the present study, and both the CG subjects and the legal guardian of each patient gave their written informed consent for study participation and data publication.

All procedures performed in studies involving human participants were in accordance with the ethical standards of the institutional and/or national research committee and with the 1964 Helsinki declaration and its later amendments or comparable ethical standards. The Institutional Review Board of IRCCS Centro Neurolesi Bonino Pulejo (Messina, Italy) approved the study. All patients provided their written informed consent to study participation and data publication.

### 2.2. Recording Session

Patients were laying in their own bed while being provided with the motor imagery tasks, whereas the control subjects were sitting on a comfortable reclining chair. Wakefulness state was ensured in patients by delivering the CRS-R arousal protocol (together with the entire CRS-R assessment) just before carrying out the tasks. We administered two types of motor imagery task, in the same day of recording, with a 60-min period of rest. The order of the two experimental sessions was random but globally counterbalanced. In the VMI, participants were watching a video showing group dances (a brief example is provided in Appendix A) on a 6-inch portable screen put in front of the patient at about 15 cm. During this task, we continuously recorded both EEG and electro-oculographic (EOG) activity. Patients’ head was held in position by a fixing device so to favor video observation. Therefore, both the distance between the screen and the face and the head-fixing device made the patients to watch the video for the time required by the experiment, also consistently with the visual scores on the subscale of the CRS-R (in the UWS sample 2 on average; Table 1). Furthermore, clinical inspection to ensure maintained vigilance was also conducted. In this regard, we sought for underarousal episodes, i.e., a contact between the upper and lower eyelids maintained continuously for more than 3 s. In such circumstance, the task was paused, the arousal facilitation protocol [51] was administered, and then the task was resumed. The EEG recording lasted 16 ± 3 min (range 13–25 min). All patients were able to carry out the task for this time span. The duration of recording depended on the frequency of underarousal; that is, the task was conducted until the underarousal episodes began to occur continuously.

They were instructed to freely observe the entire surface of the screen, without paying attention to any detail in particular. Furthermore, they had to imagine performing such a simple group dance once they heard two consecutive acoustic inputs and until they heard a stop command, or to mind wander (MW) (i.e., not to imagine dancing) once they heard one acoustic input until they heard a stop command. The participants were instructed to focus on their muscle activation as if they were truly dancing. Specifically, the subjects were provided with sequential blocks of motor imagery sessions, randomly delivered (every 5-to-10 sec). Each block was preceded by an acoustic input, i.e., a single or a double tone (square wave, duration 1 sec, 700 Hz, 70 dB, 0.5 sec of interstimulus interval), which signaled the patient to mind wandering or imagine dancing, respectively. The participant had to begin imaging dancing/mind-wandering as soon as he/she heard the acoustic signal. Each block lasted 10 sec and concluded with a stop command. The VMI and MW blocks were delivered pseudo-randomly, so no more than two blocks of the same type were completed consecutively. 

In the simple motor imagery task (simple-MI), the VMI was substituted by a task in which the participant had to imagine (not trying) to quickly squeeze the right hand into a fist and quickly relax it. EEG recording and the related procedures were the same described for the VMI-MW task. In detail, participants were instructed to imagine performing the simple hand movement once they heard two consecutive acoustic inputs and until they heard a stop command, or to MW once they heard one acoustic input until they heard a stop command. The participants were instructed to focus on their muscle activation as if they were truly squeezing the right hand into a fist. Specifically, the subjects were provided with sequential blocks of motor imagery sessions, randomly delivered (every 5-to-10 sec). The participant had to begin MI or MW as soon as he/she heard the acoustic signal. Each block lasted 10 sec and concluded with a stop command. The MI and MW blocks were delivered pseudo-randomly, so no more than two blocks of the same type were completed consecutively. 

In a control experiment, carried out in a different day, six patients in MCS, seven in UWS (see Table 1; the CRS-R scoring and the clinical condition were the same of the main experiment) and nine other CG individuals (under the same experimental conditions) were provided with a MI task in which they had to imagine group dancing (advanced-MI) and to MW; thus, without the background video. The experimental paradigm was identical to the simple-MI task. Given that the control experiment was carried out following the main experiment, we enrolled other CG participants to avoid a carryover effect of the main experiment. None of the patients was allowed for changes in medications for the whole duration of the experiments. 

### 2.3. EEG Recording 

EEG signals were recorded using a BrainQuick System (Micromed; Mogliano Veneto, Italy) equipped with a standard 21-electrode headset following the 10–20 International System. The reference and ground electrodes were placed on FCz and FPz, respectively. Electrode impedance was kept below 5 kΩ. EEG signals were sampled at 512 Hz and filtered at 0.5–45 Hz. Blinks were monitored using EOG electrodes, which were put on the horizontal line passing by the outer corners of the eyes.

In the VMI, EEG data were segmented with respect to the tone into (-1:,6) sec epochs (on average 136 ± 24, range 111–189), and baseline corrected in (-1:0) sec. After bad epoch’s rejection by visual inspection and ICA [60], we obtained 46 ± 8 VMI epochs (rejection rate of 32%) and 61 ± 6 MW epochs (rejection rate of 11%), which were used for trial analysis. We formerly found that both visual inspection and ICA were sufficient to remove ocular artifacts, notably on frontal electrodes [47,48,60,61]. We first calculated the log power values in five canonical frequency ranges (delta 1–4 Hz, theta 4–8 Hz, alpha 8–12 Hz, beta 12–30 Hz, and gamma 30–45 Hz) in the (-1:6) sec period with respect to the tone provision (which corresponded on average to the active period). Event-related synchronization/desynchronization (ERD/ERS) values were calculated according to the formula $\frac{K-R}{R}\;\times \;100$, where *K* is the average power of the active period and *R* represents the mean power of the reference period (rest); therefore, negative values corresponded to ERD, the positive ones to ERS. ERS/ERD analysis was conducted on the grouped electrodes over the left and right central areas (including C3, C4, Cz, FCz, and FPz), because the underneath areas are related to motor imagery [12], the left and right frontal areas (including F3, F4, and Fz), and the left and right parieto-occipital areas (including P3, P4, Pz, O1, and O2), because the underneath areas are related to visuomotor functions [62]. The mean topographic maps (timepoint × frequency-band interaction) showing the changes in ERS/D over time (3072 time points in each epoch) were computed for all participants. The statistical significance of the ERD/ERS values was calculated using a t-percentile bootstrap algorithm (α = 0.05) [63].

Concerning the simple-MI, the procedure was the same of the one mentioned above. Data were segmented into (-1:6) sec epochs (on average 133 ± 23, range 111–189) with respect to every tone. After bad epoch’s rejection by visual inspection and ICA, we obtained 52 ± 8 MI epochs (rejection rate of 22%) and 58 ± 6 MW epochs (rejection rate of 13%), which were used for trial analysis but limitedly to left central electrodes and alpha frequency range. 

Concerning the advanced-MI, the procedure was the same of the ones mentioned above. Data were segmented into (-1:6) sec epochs (on average 135 ± 21, range 115–191) with respect to every tone. After bad epoch’s rejection by visual inspection and ICA, we obtained 50 ± 7 advanced-MI epochs (rejection rate of 20%) and 61 ± 8 MW epochs (rejection rate of 11%), which were used for trial analysis. 

### 2.4. Statistical Analysis

Baseline comparisons concerning clinical-demographic characteristics among the groups were performed using Wilcoxon’s signed ranks test. Concerning ERS/ERD analysis, an analysis of variance (ANOVA) was used for each of the frequency bands (delta, theta, alpha, beta, and gamma for VMI task; alpha and beta for MI tasks) and task types (VMI, simple-MI, and advanced-MI) with regard to the factors “condition” (two levels: task and MW), “electrode” (six levels: left and right frontal, left and right central, and left and right parietal-occipital for VMI; the analysis in the MI tasks included only central electrode groups), and “group” (three levels: MCS, UWS, and CG). A *p*-value < 0.05 was considered significant. Correlation analysis was carried out between clinical and electrophysiological data using Pearson or point-biserial correlation tests, where appropriate. 

The accuracy of the ERS/D in predicting group membership was estimated as the number of correct predictions (each ERS/ERD) from all predictions made (all EEG trials). In this regard, the logarithmic band power for each frequency band was calculated in consecutive, non-overlapping 1-sec sample-by-sample in each trial (by band-pass filtering, squaring and averaging the samples) compared to a reference (i.e., 1 s before the command delivery). A single value was sampled at the middle of each time-period, referenced to the baseline period, and was thus used in the classification accuracy analysis among all frequencies and electrode groups. To this end, we used a linear discriminant analysis classifier, with a nested block-wise cross-validation (10 × 10 inner fold; leave-one-out-block outer fold) being applied (to avoid overfitting) [12,64,65]. Classification accuracies were compared using the reliable change index at individual level and *t*-tests at group level (*p*ost hoc analysis) among all electrode groups and frequency bands as estimated with the block-wise nested cross-validation [65]. The experimenter who analyzed the EEG data was blind to patients’ behavioral responsiveness level.

## 3. Results

There were no significant differences between patients with UWS and MCS concerning clinical-demographic characteristics, but CRS-R as foreseeable (Table 1). 

VMI induced distinct ERS/ERD in CG and MCS group within each electrode group and in all frequency bands, but delta, as indicated by the significant interactions and effects among the factors condition, electrode, and group (Table 2). In particular, both CG and MCS showed an alpha and beta ERD at left central electrode group (CG *p* < 0.001, MCS *p* = 0.01) (*p*aralleled by a non-significant alpha and beta ERS within right central electrode group; side comparison *p* < 0.001), a theta and gamma ERS at both frontal electrode groups (both *p* < 0.001; side comparison *p* > 0.1), and a theta and gamma ERD at both parieto-occipital electrode groups (both *p* < 0.001; side comparison *p* > 0.1) in response to the VMI (Figure 1), which were all significantly different between these two groups in terms of response magnitude (*p* = 0.01). Contrarily, none of the patients in UWS, but one (n.8) showed ERS/ERD response to the task (Figure 1) (all CG-UWS and MCS-UWS comparison *p* < 0.001). Particularly, the UWS patient no. 8 was a 33-year-old man who suffered nine months before of a severe traumatic brain injury after car accident with diffuse axonal injury and a frontal hematoma (that was evacuated). He was admitted to our unit about four months after brain injury. His CRS-R score was of three (auditory startle, visual startle, tetraplegia, no oromotor/verbal function, eye opening with stimulation, and no communication). At ninth month post-injury, he achieved a CRS-R score of six (auditory startle, fixation, tetraplegia, oral reflexive movements, eye opening without stimulation, and no communication). During the hospitalization, he was provided with usual cares (skin and oral care, and management of bowel and bladder functions), intensive conventional physiotherapy at the bedside, levodopa equivalent daily dose of 750 mg, antispastic (intrathecal baclofen pump at 50 µg/day), and painkillers (fentanyl transdermal patch 50 µg/hour). He breathed spontaneously through tracheostomy and was fed through gastrostomy tube. At the EEG analysis, he showed responses resembling those showed by patients in MCS, although with smaller magnitude and the lack of beta ERD (being alpha ERD preserved) (Figure 1). MW returned no ERS/ERD response to the tasks in all groups (all *p* > 0.1) (Figure 1). Condition comparison (VMI vs. MW) was always significant in CG and MCS (*p* < 0.001) but not in UWS (*p* > 0.1).

The simple-MI in CG and MCS group induced an ERS in left central electrode group in alpha and beta frequency bands (Figure 1; Table 2). Contrarily, UWS showed no ERS/ERD response to the task (Figure 1) (all CG-UWS and MCS-UWS comparison *p* < 0.001). The responses were significantly different between CG and MCS groups in terms of magnitude (*p* = 0.01). MW returned no ERS/ERD response to the tasks in all groups (all *p* > 0.1). Condition comparison (simple-MI vs. MW) was always significant in CG and MCS (*p* < 0.001) but not in UWS (*p* > 0.1).

Lastly, the advanced-MI in CG and MCS group induced an alpha ERD prominently in left central electrode group and beta ERD in both central electrode groups (Figure 1; Table 2). Contrarily, patients in UWS showed no ERS/ERD response to the task (Figure 1), but the patient in UWS no. 8 (only alpha ERD) (all CG-UWS and MCS-UWS comparison *p* < 0.001). The responses were significantly different between CG and MCS groups in terms of magnitude (*p* = 0.01). MW returned no ERS/ERD response to the tasks in all groups (all *p* > 0.1). Condition comparison (advanced-MI vs. MW) was always significant in CG and MCS (*p* < 0.001) but not in UWS (*p* > 0.1).

The findings from the CG confirm that the tasks we employed furnished the expected responses (Table 1). Actually, all CG individuals produced significant EEG responses consistent with the VMI task (i.e., they were able to modulate their brain activity by generating voluntary, reliable, and repeatable EEG responses in predefined neuroanatomical regions), with a classification accuracy of 89 ± 7% (range 80–95%). On the contrary, six individuals (73%) showed EEG responses consistent with the advanced-MI task, with a classification accuracy of 72 ± 9% (range 61–80%). Lastly, 10 participants (66%) showed a lower classification accuracy in the simple-MI task (89 ± 11% range 60–70%). The MW returned no EEG responses classifiable as expression of command following (mean 27%, range 19–35%; all *p* > 0.2). When comparing tasks outcomes, the VMI largely outperformed both MI tasks at both individual (all *p* < 0.01) and group level (both *p* < 0.001); in addition, the aMI outperformed the sMI task, although this did not achieve the statistical significance.

All patients who were labeled as MCS showed positive EEG responses to the VMI task. However, these responses were significant in six patients (66%). The classification accuracies for these patients was 77 ± 16% (range 50–93%). None of the EEG recordings for these patients were significant during the MW condition (mean 23%; all *p* > 0.2). Consistently, scalp map showed a significant distribution of the ERD/S (timepoint × frequency-band interaction) within specific electrode groups, with clear and distinct foci over the sensorimotor and fronto-parieto-occipital areas that were formally identical in CG and MCS groups (who thus significantly followed commands in this task) (Figure 1). On the contrary, four individuals (44%) showed EEG responses consistent with the simple-MI task, with a lower classification accuracy (51 ± 13%, range 37–64%). Lastly, three patients (55%) showed a lower classification accuracy in the advanced-MI task (50 ± 5% range 43–56%). The MW returned no EEG responses classifiable as expression of command following (all *p* > 0.1). When comparing tasks outcomes, the VMI largely outperformed both MI tasks at both individual and group level (Table 1); in addition, the aMI outperformed the sMI task, although this did not achieve the statistical significance.

Contrarily, none of the patients with UWS, but one (no. 8), showed any appreciable EEG sign in response to both tasks (all *p* > 0.1) (Figure 1; Table 1). Actually, the classification accuracy was significantly low in all patients (Table 1). None of the EEG recordings for these patients were significant during the MW condition (mean 21%; all *p* > 0.2). Specifically, scalp map did not show any significant distribution of the ERD/S (timepoint × frequency-band interaction) within specific electrode groups, with no clear foci (suggesting that these patients did not follow commands with this EEG task) (Figure 1). Notwithstanding, a patient in UWS (no. 8) showed a clear frontoparietal theta/gamma ERS but a very mild alpha but not beta ERD (Figure 1), with a classification accuracy that was similar to that showed by the MCS sample (Table 1). This patient showed a reliable response to the advanced-MI task but not to the simple-MI task (Figure 1; Table 1).

None of the clinical-demographic features (including age at time of injury; time since injury, CRS-R score, and etiology) significantly affected both EEG findings and the prediction of patient’s ability to follow commands in this EEG task.

## 4. Discussion

All CG individuals, all patients in MCS, and one patient in UWS showed specific EEG responses to the diverse motor imagery tasks, which were consistent with motor imagery processes. Therefore, the present data agree with the usefulness and reliability of EEG in detecting awareness [4,12,13,14,15,16,17,18,19,40,66,67]. Furthermore, our study suggests that EEG responses to the VMI could be reliable in differentiating DoC patients, as suggested by the high classification accuracy scores that clearly outperformed those of MI tasks. Therefore, the significant novelty of our approach consisted in the fact that it was (i) visuomotor-assisted, thus providing patients with more entraining and cognitive-targeting stimuli, rather than adopting standard and pragmatic motor tasks (as in MI); and (ii) considerably cheaper (21 EEG channels compared to high-density EEG) and easily applicable at the bedside. Notably, this approach identified one patient in UWS who seemed to be aware according to our data, i.e., he was able to substantially modulate his EEG responses and consistently to command. Therefore, the present approach could bring to light covert awareness in patients seemingly UWS. This is of significant importance, given that the standard clinical assessments of command following based on behavioral observation can be biased by multiple factors depending on the examiner (clinical assessment is fundamentally subjective) and the patient (*p*oor cooperation, sensory and motor deficits, and cognitive impairment) [14].

### 4.1. Reliability of the EEG Changes in Terms of Motor Imagery

The reliability of the present EEG changes in terms of motor imagery is demonstrated by the fact that there was a significant overlap among the EEG responses to the VMI, which are known to occur in aware individuals [4,12,13,14,15,16,17,18,19,40,66,67], in all MCS, all CG, and one UWS patient. Furthermore, these EEG modulations were largely consistent across the tasks within the same imagery type (namely, a significant timepoint × frequency-band interaction), but also largely differed among VMI and MI tasks, thus suggesting the reliability of the EEG changes we described in terms of volitional EEG modulation by part of the patients. Specifically, the alpha and beta ERD within central electrodes may reflect the motor output planning, whereas the theta/gamma ERS/ERD within frontal and parieto-occipital electrodes suggests the activation of visuomotor pathways related to motor programming and planning in response to the visuomotor task [61,68,69,70,71,72,73,74]. Notably, alpha and beta EEG responses to VMI task within central electrodes were lateralized to left hemisphere. This different result as compared to the literature data [75] may depend on several issues, including the different types of motor imagery, extent of movement duration, feedback interference of muscle network activation during motor imagery task, activation of subcortical structures (that are critically damaged in DoC, especially post-anoxic) [76], and motor-cognitive disability degree [75,77,78]. Furthermore, it is likely that the mirror neuron system (MNS) largely contributed to generate these EEG responses, as we previously demonstrated in patients with DoC [79], further substantiating the reason why VMI outperformed MI. Even though the changes in corticomotor excitability during action observation (alpha/beta ERS/ERD) is the established and accepted feature of MNS activation, and the timing and lateralization of the EEG responses expressing the potential contribution of MNS to motor imagery is yet obscure, the plasticity of the MNS (theta/gamma ERS/ERD) could have resulted in a lastingly and lateralized increased corticomotor excitability that was greater during VMI than during MI [80,81,82]. Actually, MNS activation may occur owing to the task we adopted (specific, goal-directed motor act) [83]. Furthermore, MNS activation may cause a mental reiteration of the motor task just performed even during MW, even in an unconscious manner [84]. 

On the contrary, alpha and beta EEG changes in response to sustained MI tasks (alpha and beta ERS) [75,77,78] were focused on left central electrodes, as foreseeable based on the EEG analysis we applied and the literature data [78,85,86,87,88]. Moreover, these responses were appreciable in both CG and MCS participants but not in UWS persons. Consistently with these findings and the literature data, these changes express motor preparation and execution mechanisms over time [89,90,91] at an aware level [12,26,33,64]. 

Lastly, we compared the condition of imaging the group dance with the background video (i.e., VMI task), with the condition of imaging a group dance without a background video (advanced-MI). This was carried out to confirm the usefulness of providing patients with DoC with video feedback during motor imagery trials to bring to light awareness. Actually, the advanced-MI task yielded EEG responses consistent with motor imagery (bilateral alpha and beta ERD) only in a portion of CG individuals, in a few patients with MCS, and in none of those with UWS. These responses were even stronger and more distributed than those concerning simple-MI tasks were [70]. This is consistent with the issue that more vivid and familiar tasks not only drives comparable cortical networks to those for a simple task, but also that they are more effective [92,93,94], and that the sensorimotor ERS/ERD quantifying the activity of the brain during motor imagery tasks is influenced by the selection of the optimal baseline (instructive video) [95]. Lastly, complex but common mental tasks can enhance the single-trial detectability of imagined movements [32]. This may explain the reason why advanced-MI task was not accompanied by fronto-parieto-occipital theta and gamma ERS/ERD, which instead characterized VMI task. The reason why simple-MI tasks yielded ERS responses whereas VMI and advanced-MI tasks returned ERD responses to prolonged motor imagery may likely depend on several issues, including the different degree of mental effort, the video provision, the ongoing cortical activity, the personal previous motoric experiences, and the different extent of muscle network activation required [77,86,87,96,97]. Conversely, all HC and MCS participants showed EEG responses to VMI task consistent with motor imagery processes. Therefore, our result indicate that EEG responses came from the video provision and not because the task of performing group dance was easier to imagine than squeezing a hand. Given that this type of motor imagery expresses the mental rehearsal of complex actions without engaging in the actual movements involved, there is suggestion for awareness preservation in case of positive response.

### 4.2. Significance of EEG Responses to Motor Imagery Task toward Awareness

One could argue that the EEG responses may be biased by something else that covert awareness. For instance, EEG modulation may be biased by the visual feedback itself, which could mask, or even substitute, the genuine EEG responses to the motor imagery command. In other words, the present EEG findings may represent an effect of sustained movement observation rather than of motor imagery. Four issues contradict these concerns in our opinion. First, none of the patients and controls showed specific EEG responses to MW. Therefore, EEG responses are specific to different tasks, reflecting a willful, top-down information processing, e.g., the capability of an individual to rehearse the selection of stored visual representations in response to proper cues (visual stimulation) [98]. Second, the EEG responses were all specific for the task we employed, including alpha/beta ERD within central electrodes and theta/gamma ERS within frontal and parieto-occipital electrodes, consistently with the abovementioned top-down information processing [98]. Third, specific EEG activities before command delivering were not appreciable in all participants. This further suggests that all responsive participants produced task-appropriate EEG responses in due time with the cues, as expected. Last, we observed specific changes in distinct frequency ranges, which makes unlikely a random effect of the tasks we adopted on willful EEG modulation, but rather suggests a precise cognitive entrainment. Whole band frequency perturbation is indeed in line with our previous work on visuomotor processes in patients with DoC [47,48,99].

Furthermore, EEG responses may be biased by an automatic rather than conscious and overt response to the acoustic stimuli that were delivered with video provision and the signals to the patient when starting to imagine dancing or mind wandering. Even though the video could have been silent to avoid further auditory stimulation, we opted for a musical video to favor cognitive entrainment and the motor imagery execution. Furthermore, temporal electrodes were not included in the analysis (even though signals may spread from temporal areas to the other regions). Thus, whether a silent or musical video will result in better effects will be addressed in other studies. It is true that patients had to imagine dancing when they heard a double tone and to MW when they heard a single tone. Thus, the EEG differences between the tasks may originate from this acoustic difference. However, an analogue difference was appreciable when providing patients with MI tasks and, above all, we got totally different EEG modulations, i.e., largely consistent within the task and largely different among the tasks. Furthermore, no CG and MCS participants returned any EEG response to the MW condition. Lastly, if the acoustic stimuli had generated these responses, they should have had to abate and recur in synchrony with these cues. All these issues suggest that listening to the instructions was sufficient to generate neither automatic brain responses nor task-related EEG changes, and that the EEG changes we described were reliable in terms of volitional EEG modulation by part of the patients. Consequently, the fact that the listening to the instructions may have been sufficient to mimic task performance is unlikely. However, to further confirm this issue, it could be useful to counterbalance the trials with a double tone, half being associated to VMI, and half to MW in a next, randomized clinical trial.

One may wonder whether the EEG activations we reported on were expression of a non-specific degradation of EEG connectivity and oscillatory activity rather than of top-down sensorimotor processing functions in patients with DoC. Further, EEG responses may not necessarily reflect willful EEG modulation in keeping with the issue that regular movement observation also reaches eventually the awareness level [100,101]. However, the EEG activations were specific for different motor tasks both in DoC patients and the CG participants, consistently with the literature data [102,103,104], thus suggesting the selective top-down modulation from high-order frontal to primary sensory-motor areas, which implies aware motor control (*p*rogramming and planning) even in the case of regular presentation of movements [105,106,107,108]. Therefore, any EEG response consistent with the VMI in terms of frequency range (alpha/beta and theta/gamma), scalp distribution (central and frontal-parietal electrodes, respectively), and reliability over time (timepoint × frequency-band interaction) may underline an aware cognitive process. 

The content of the activated cognitive process by means of the present task is not definable with the present data. However, we can hypothesize that many cognitive functions had to be entrained in order to carry out and complete the task, including sustained attention, response selection (between the tasks), language comprehension (of the task instructions), and working memory (to remember which task to do). This cognitive entrainment may also explain the fact that we observed changes in many frequency ranges. Whole band frequency perturbation is indeed in line with our previous work on visuomotor processes in patients with DoC [47,48,99]. As the issue of testing visuomotor processes with these types of approaches is recent, we can only make a hypothesis on the whole frequency range perturbation. This may reflect a hyper-facilitated brain responsiveness secondary to severe brain damage [109,110,111] in the attempt to recover brain function in the framework of a large-scale connectivity breakdown that “stresses” the brain to work in many frequency ranges, as proposed by very recent data on multiplex–multilayer network analyses [99]. Anyway, all these cognitive processes reflect a top-down cognitive control that, in turn, means awareness [112]. Concerning working memory, one may argue that a limitation of this cognitive function may cause a limited, short lasting, and thus non-appreciable response. However, this was the case of all patients with UWS, but one, and none of MCS and CG, making this concern unreliable. 

It has been shown that healthy individuals cannot necessarily show EEG responses to MI task training (so-called brain–computer interface illiterates) [113], whereas all of them showed significant EEG responses to VMI. Furthermore, there could be the circumstance that a patient with DoC does not show significant EEG response to a motor imagery task for a bulk of possible reasons, including low sensitivity of the method, patients’ limited residual cognitive capabilities, transient unawareness during the experimental paradigm (particularly for MCS and covert awareness cases), deficits in language comprehension, working memory, decision making, or executive function, which are all known factors biasing willful modulation experimental tasks in both EEG and functional neuroimaging studies. Actually, it is challenging to verify whether the negative result depends on one or more among the abovementioned biasing factors. Therefore, it is preferable to emphasize only the positive results, given that they confirm the preservation of aware sensorimotor, top-down processes, regardless of corroborative behavioral data [79,83].

We did not find any effect of clinical-demographic characteristics, including medications, on EEG data, which may be surprising unless we consider that all patients were tested in the same condition (non-sedated, after having received the CRS-R arousal protocol). Nonetheless, the effects of the different combinations of drugs used in severely brain injured and bedridden patients on the magnitude of task-induced EEG changes remain unclear and, therefore, deserve further studies [114]. Furthermore, it has been reported that etiology correlates with behavioral responsiveness, particularly in the long-term, in the UWS but not MCS patients [115]. The discrepancy between our and literature data on the well-known effects of age, gender, disease duration, and initial CRS-R score on clinical-electrophysiological correlations may depend on the fact that age was rather variable, disease duration was restricted by the inclusion criteria and was higher in MCS (smaller sample) than in UWS group. As we were interested on between-group differences, some peculiar characteristics may have been lost in the analysis, which should be addressed in further clinical trial.

Why some patients in UWS are covertly aware is still debated. Our data point out that the dissociation between movement execution (impaired) and motor imagery networks (preserved) that characterized a patient with UWS in our study further suggests that motor output failure may be responsible for behavioral unresponsiveness in these patients with UWS. Indeed, the “covertly” aware patients show a clear dissociation between imagining (motor imagery) and performing (motor execution) a motor action. Although, a patient who is capable of motor imagery should also be capable of motor execution. This motor output dissociation can be related to the failure of motor thalamo-cortical circuits (from the thalamus to M1 or SMA), which have been proposed as possible biomarker for the absence of intentional movement in covertly aware patients [116]. These persons should be therefore more properly diagnosed as non-behavioral MCS, cognitive-motor dissociation, or functional locked-in syndrome [4,12,13,14,15,16,17,18,19,21,21,40,66,67,117].

## 5. Conclusions

Our data provide a proof of concept for the possibility of applying VMI as an aid in the differential diagnosis of patients with DoC and, possibly, in identifying the covertly aware patients, despite we used a cheaper and more accessible method than the currently available advanced assessment techniques. However, our study has some limitations to acknowledge. First, the small sample size and the lack of other control tasks than MW. Obviously, as this is a proof of concept study, further clinical trials have to confirm our findings on a large scale. Second, as already stated above, the lack of EEG response may be due to a natural, non-specific degradation of EEG oscillations rather than to a specific cognitive function impairment. Nonetheless, the EEG activations we described were task-specific, consistently with former evidences [118,119,120,121,122]. Lastly, the well-known lack of fixation and eye tracking in patients with UWS could limit movement observation and execution tasks, although all patients were treated with l-DOPA, which has well-known positive effects on pursuit ocular movements [123]. Consequently, the comparison between patients’ and controls’ EEG may be deemed as problematic. However, brain oscillations were clearly different between the groups, and highlighted aware brain activity even in the absence of motor behavior. Furthermore, the experimental setup guaranteed an adequate observation of the motor scenario, with no discomfort.

Therefore, this method may be considered as, at least, a screening tool to identify the patients who deserve further, advanced paraclinical approaches (neurophysiologic and functional neuroimaging). Furthermore, it is possible to suggest that motor imagery tasks should be delivered to the patients in visuomotor-aided modality, also in the rehabilitation setting, to greatly entrain patient’s participation, regardless of online feedback. Lastly, our approach may pave the way for the development of brain–computer interfaces or non-invasive brain stimulation paradigms aimed at improving communication and awareness recovery.

## Figures and Tables

**Figure 1 brainsci-10-00746-f001:**
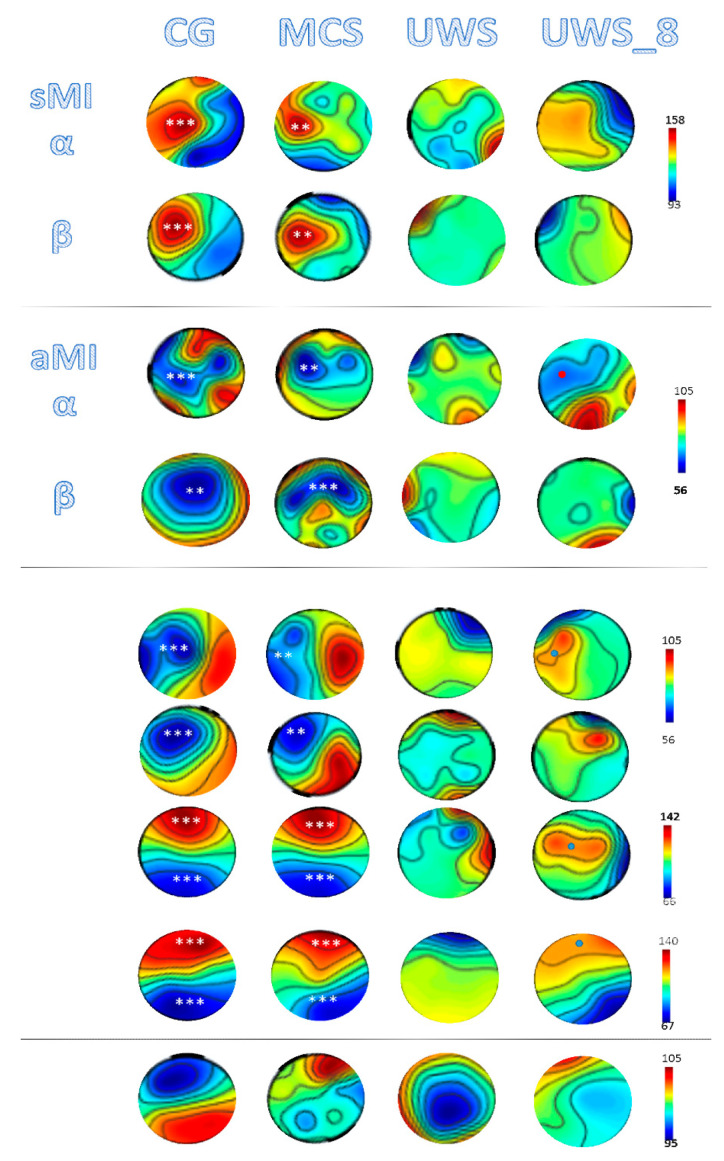
Event-related synchronization/desynchronization (ERD/ERS) grand average topographies within the appropriate frequency bands, which are associated with visuomotor (VMI), simple (sMI), advanced motor imagery (aMI) tasks, and mind wandering (MW) in each group and the unresponsive wakefulness syndrome (UWS) subject no. 8. Maps show the scalp distribution of the ERS/ERD (time-point × frequency-band). Red colors show% power changes greater than baseline, blue colors smaller than baseline. The EEG outcomes, i.e., clear foci over the sensorimotor areas, were similar in topography but of greater amplitude in control group (CG) compared to minimally conscious state (MCS) participant (*p* < 0.001), and the UWS patient n.8 (dots), meaning that all these subjects significantly followed commands with these EEG tasks. Statistical comparison (timepoint × frequency-band) of grouped electrodes (*** *p* < 0.001, ** *p* < 0.01) for CG vs. MCS and MCS vs. UWS are shown in the CG and the MCS column, respectively (CG vs. UWS was always *p* < 0.001, not shown). The color bars refer to% changes of the log power spectral densities ($10\;\times \;log_{10}\left(\frac{\mu V^{2}}{Hz}\right)$).

**Table 1 brainsci-10-00746-t001:** Shows the clinical demographic characteristics and the features of electroencephalography (EEG) during the visuomotor imagery (VMI), simple (sMI), and advanced motor imagery tasks (aMI) (EEG classification accuracy (CA), %, and *p*-value for EEG command following; for the post hoc analysis, CA were compared at individual level using the reliable change index, and at a group level by *t*-tests). The possibly misdiagnosed patient with UWS (no.8) is highlighted with *. Legend: DAI diffuse axonal injury, WMH white matter hyperintensity, F frontal, Fb frontobasal, FP frontoparietal, _h hemorrhagic lesion, _is ischemia, SAH subarachnoid hemorrhage, TPO temporo-parieto-occipital, dd disease duration, T traumatic, V vascular, A anoxic, na not applicable, np not performed.

DoC	Gender (F:M)	Age (y)	dd (m)	Etiology	MRI	CRS-R	Visual Function	VMI		sMI		aMI		VMI-sMI	VMI-aMI	sMI-aMI
								CA	*p*	CA	*p*	CA	p			
MCS (*n* = 9)	F	72	26	T	multiple_h	14	4	30	0.01	21	0.2	np		1.1	na	na
	M	74	17	A	WMH	14	4	60	0.005	41	0.01	26	0.1	2.3	4	2.5
	M	35	23	A	WMH	19	5	50	0.02	37	0.01	np		1.5	na	na
	F	43	19	T	DAI Fb_h	14	3	92	0.006	16	0.3	60	0.04	9	3.8	7.4
	F	35	18	A	WMH	17	4	70	0.03	27	0.2	28	0.1	5.1	5.1	0.2
	M	50	16	T	multiple_h	18	5	93	0.1	64	0.01	21	0.2	3.4	8.5	7.4
	F	51	15	V	DAI Fb_h	14	4	75	0.1	26	0.2	41	0.04	5.8	4	2.5
	M	36	13	V	SAH	14	3	80	0.002	60	0.01	50	0.01	2.3	3.5	1.8
	F	32	12	A	WMH	14	4	95	0.1	22	0.2	np		8.6	na	na
mean ± sd	5:4	48 ± 16	18 ± 5			15 ± 2	4 ± 0.7	72 ± 22		35 ± 17				0.001	0.006	0.8
UWS (*n* = 11)	M	56	27	V	SAH	7	2	29	0.1	24	0.2	23	0.1	0.1	1	0.2
	F	50	16	A	WMH	7	1	21	0.2	25	0.2	np		0.1	na	na
	F	40	20	T	DAI F_h	7	2	28	0.2	12	0.3	23	0.1	0.4	0.8	1.2
	F	78	11	V	FP_is	5	2	23	0.1	12	0.3	np		0.2	na	na
	F	47	10	T	DAI FP_h	5	1	49	0.1	23	0.2	np		0.6	na	na
	M	46	17	T	DAI F_h	5	2	38	0.1	21	0.2	27	0.1	0.4	1.8	1.2
	M	27	12	V	SAH	7	2	28	0.1	19	0.2	29	0.1	0.2	0.2	1
	M*	33	9	T	DAI F_h	6	2	81	0.03	59	0.2	70	0.04	3.3	2.6	2.1
	F	23	10	A	WMH	7	2	35	0.1	21	0.1	25	0.1	0.5	1.9	0.9
	F	39	15	A	WMH	7	2	30	0.1	19	0.2	np		0.3	na	na
	M	77	14	T	TPO_h	7	2	26	0.1	24	0.2	25	0.1	0.1	0.2	0.2
mean ± sd	6:5	47 ± 18	15 ± 5			6 ± 1	2 ± 0.3	36 ± 20		24 ± 15				0.1	0.6	0.4
*p*-value	0.97	0.9	0.2	0.7	0.5	<0.001	<0.001	0.001		0.2				0.001	0.001	0.8

**Table 2 brainsci-10-00746-t002:** ANOVA interactions for ERS/D modulations in each band, group, and electrode group during motor tasks. Delta frequency range was not significant (not shown). np not performed.

Task	Interaction	Theta (F p)		Alpha (F p)		Beta (F p)		Gamma (F p)	
VMI	condition × group F_(2 64)_	0.08		32	<0.0001	26	<0.0001	0.1	
	electrode × group F_(10 320)_	40	<0.0001	34	<0.0001	22	<0.0001	43	<0.0001
	condition × electrode F_(5 160)_	22	<0.0001	66	<0.0001	65	<0.0001	22	<0.0001
	condition × group × electrode F_(10 320)_	37	<0.0001	28	<0.0001	25	<0.0001	40	<0.0001
simple-MI	condition × group F_(2 64)_	np		8.2	0.0004	9.2	0.0003	np	
advanced-MI	condition × group F_(2 38)_	>0.1		11	<0.0001	11	<0.0001	>0.1	
	electrode × group F_(10 190)_	>0.1		4.1	<0.0001	3.6	0.0002	>0.1	
	condition × electrode F_(5 95)_	>0.1		7.6	<0.0001	6.7	<0.0001	>0.1	
	condition × group × electrode F_(10 190)_	>0.1		4.6	<0.0001	4.7	<0.0001	>0.1	

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
