# Peer review of "Towards New Diagnostic Approaches in Disorders of Consciousness: A Proof of Concept Study on the Promising Use of Imagery Visuomotor Task"

_brainsci, 2020, doi:10.3390/brainsci10100746_

Round 1
Reviewer 1 Report
This is an interesting study focusing on the development and validation of a new paradigm to assess covert cognition in DOC patients using a visuomotor guided motor imagery task during EEG recording (one single recording session). The paper has clear clinical, ethical and legal relevance since there is growing scientific interest in studying those patients that may be aware despite their inability to manifest it behaviorally.
Previous studies have shown the utility of EEG in improving the classification accuracy of DOC, especially in patients who are not able to express conscious responses in behavioral assessment. However, the variability and complexity of EEG measures and analysis, create obstacles for becoming a practical and widely applicable tool. Here, the authors combine the novelty of presenting a visuomotor-guided task (most previous active paradigms are based on auditory stimulus) with the added value of using a simple 21-channels EEG device, which may facilitate its application in DOC clinics. Although the research is presented as a proof of concept study, results presented here are consistent with previous literature, reinforcing the idea that EEG can be used as a useful tool to assess the preservation of functional brain networks in DOC population.
In my opinion, these points render this submission as written accepted for publication considering only minor edit changes that should be discussed/emphasized or clarified.
Introduction:
Please review p2 line 68: “ In this ? wake…..
If possible, discuss (here or in the discussion) previous findings and limitations of previous studies focused on the use of EEG to the detect cover awareness in DOC patients (remark the novelty of your paradigm and the benefits of a 21ch device)
Materials
P-3 l-95: “The awareness level was assessed for one month using the CRS-r”
Please could you be more precise? (eg: periodicity, same evaluator? etc.); and also, if possible, please could you define the interval between last DOC assessment and active task?. Finally, please clarify if changes in medications were or were not allowed.
Table 1
“Tasks no” does not add relevant info and complicates the legibility of the table. Please include etiology in the legend. Please check the patient highlighted in bold (in my version is a MCS patient)
Please review p4 line 136: “squeeze the right ? into a fist…
Results
The information is well presented but, since cover cognition was detected in only one UWS patient of the total sample, it would be great if the authors can give some more additional info regarding this specific patient (eg: CRS-R subscale scores, medications, or any other significant clinical data that might be correlated with recovery in DOC patients).
Discussion
- The discussion is well organized and sustains the main findings of the paper
- Page 7 Line 232-235: Could you clarify the term: "entraining"?
- Please define the meaning of MNS in the text (mirror neuron system)
- Please review that ref 47 is consistent with the text: "Actually, MNS activation may occur owing to the task we adopted (specific, goal-directed motor act) [47]
- Please review p8 line 296; “characterized two ? among…..
- Please review the discussion about the neurobiological substrate of covert cognition (p 8 line: 294-300) since cortico-thalamic connections (from the thalamus to M1 or SMA) have been proposed as possible biomarker for the absence of intentional movement in covertly aware patients.
Author Response
Dear Editor,
We want to thank you and your reviewers for the appreciation to our manuscript and the useful comments to improve its quality. We have addressed reviewers’ concerns in the present R1 response letter. Changes in the manuscript were highlighted using red characters.
To the Reviewer #1:
- This is an interesting study focusing on the development and validation of a new paradigm to assess covert cognition in DOC patients using a visuomotor guided motor imagery task during EEG recording (one single recording session). The paper has clear clinical, ethical and legal relevance since there is growing scientific interest in studying those patients that may be aware despite their inability to manifest it behaviorally. Previous studies have shown the utility of EEG in improving the classification accuracy of DOC, especially in patients who are not able to express conscious responses in behavioral assessment. However, the variability and complexity of EEG measures and analysis, create obstacles for becoming a practical and widely applicable tool. Here, the authors combine the novelty of presenting a visuomotor-guided task (most previous active paradigms are based on auditory stimulus) with the added value of using a simple 21-channels EEG device, which may facilitate its application in DOC clinics. Although the research is presented as a proof of concept study, results presented here are consistent with previous literature, reinforcing the idea that EEG can be used as a useful tool to assess the preservation of functional brain networks in DOC population.In my opinion, these points render this submission as written accepted for publication considering only minor edit changes that should be discussed/emphasized or clarified.
We want to thank the reviewer for the appreciation to our manuscript and the useful comments to improve its quality.
- Introduction:
- Please review p2 line 68: “ In this ? wake…..
Checked and corrected in “Particularly”
- If possible, discuss (here or in the discussion) previous findings and limitations of previous studies focused on the use of EEG to the detect cover awareness in DOC patients (remark the novelty of your paradigm and the benefits of a 21ch device).
Thank you for your suggestion. Accordingly, we added a paragraph in the introduction on this issue.
- Materials
- P-3 l-95: “The awareness level was assessed for one month using the CRS-r” Please could you be more precise? (eg: periodicity, same evaluator? etc.); and also, if possible, please could you define the interval between last DOC assessment and active task?. Finally, please clarify if changes in medications were or were not allowed.
We specified that the CRS-R was administered every three days by two assessors expert in CRS-R administration. Furthermore, the CRS-R arousal protocol was delivered just before carrying out the tasks together with the entire CRS-R assessment. None of the patients was allowed for changes in medications for the whole duration of the experiments, as now better specified.
- Table 1: “Tasks no” does not add relevant info and complicates the legibility of the table. Please include etiology in the legend. Please check the patient highlighted in bold (in my version is a MCS patient)
Table 1 was checked and corrected as suggested
- Please review p4 line 136: “squeeze the right ? into a fist…
Checked and corrected (hand)
- Results
- The information is well presented but, since cover cognition was detected in only one UWS patient of the total sample, it would be great if the authors can give some more additional info regarding this specific patient (eg: CRS-R subscale scores, medications, or any other significant clinical data that might be correlated with recovery in DOC patients).
Thank you for your appreciation. As suggested, we added more specific information on this patient. It is in fact true that post-traumatic DoC has significant chances to recover compared to non-traumatic patients, which may explain why the patient in UWS n.8 succeeded in performing the tasks. However, other two patients in UWS with a similar disease duration and etiology did not show any positive response to the VM tasks. Further, clinical and demographic features were rather homogeneous among patients and they did not affect EEG findings in our correlation analysis. Therefore, further larger sample studies may clarify this interesting issue.
- Discussion
- The discussion is well organized and sustains the main findings of the paper.
Thank you again for your appreciation.
- - Page 7 Line 232-235: Could you clarify the term: "entraining"?
Checked and corrected in “captivating”
- Please define the meaning of MNS in the text (mirror neuron system)
Checked and corrected
- Please review that ref 47 is consistent with the text: "Actually, MNS activation may occur owing to the task we adopted (specific, goal-directed motor act) [47]
Checked and corrected
- Please review p8 line 296; “characterized two ? among…..
Checked and corrected in “a patient with UWS
- Please review the discussion about the neurobiological substrate of covert cognition (p 8 line: 294-300) since cortico-thalamic connections (from the thalamus to M1 or SMA) have been proposed as possible biomarker for the absence of intentional movement in covertly aware patients.
We thank the reviewer for this important information that was added in the discussion.
Kindest regards,
The authors
Reviewer 2 Report
This manuscript offers a new approach in the detection of covert awareness in patients with DoCs. Authors used a visuo-motor guided imagery task in 20 patients with DoCs and found that all patients with MCS generated appropriate EEG responses, as well as one patient with an unresponsive Wakefulness Syndrome (UWS). Compared to a classical motor imagery paradigm, visuo-motor guided imagery provided much better results, suggesting that this method could be used as a screening tool for patients with DoCs.
First of all, I have to say that I found this article very well written. Ideas are clearly exposed and as a reader it is easy to follow the intentions of the authors. I want to congratulate the authors for this. Moreover, I found that the patients sample not so small. I know how difficult it is to recruit such patients.
However, there are major concerns that stroke me throughout the manuscript and make me wonder about the validity of the results presented here.
The results are very impressive. Despite the task being rather difficult for a patient, all patients (including MCS and UWS) had p-values below <0.1 both in the IVM task AND the MI task (table 1). I don’t know if the Bonferroni correction has any sense here since this is an exploratory study and data are analyzed at the individual level. The fact that these results can be observed in (almost) all frequency ranges and all conditions is also very surprising. The absence of relationship with clinical characteristics may also appear surprising. At least, most of UWS subjects should have had p value far above 0.1 in the MI task.
I don’t blame the authors for obtaining so impressive results, but necessarily one has to remove all other potential explanations that could explain why this worked so well (and also to replicate the study).
Of course, the main limitation I see, and that is recognized by the authors, concerns the lack of fixation and eye-tracking of patients with DoCs. In fact, from my own experience, I don’t know how possible it is to make these patients watch a video for such a long time (which does not mean it is impossible). Consequently, could the authors explain in more details what the sentence l.132-133: “Clinical inspection to ensure maintained vigilance was also conducted.” Actually mean? What did they do if the patient did not maintain vigilance?
Moreover, did the ocular artifacts were sufficiently removed after bad epoch rejection and ICA? Notably on frontal electrodes that were removed for analysis. The authors write l.145 that they obtained 70 +/- 14 epochs. Was it for IVM only or for IVM + MW? What was the percentage of rejected epochs?
To finish on the question of eye movements, controls were on a chair, whereas patients were lying on a bed. Is it sure that patients (especially UWS) could efficiently watch the video? Was their head raised by some pillows for example?
When looking at figure 1, I am surprised that only the control group activated the occipital region (and only during MW). Shouldn’t we expect to see this region activated in all subjects while watching a movie?
Maybe the authors should also report the visual scores on the subscale of the CRS-R to ensure that they were able to follow a visual stimulation.
My second main concerns are related to the design of the study. How can we be sure that EEG responses are not biased by something else that covert awareness. There is a paragraph about this in the discussion but I don’t think this is enough.
For example, isn’t it possible that the different EEG responses are not reflecting any perceptual process? In the IVM task they have to imagine dancing when they hear a double tone, whereas in the MW they only hear a single tone. May the signal differences originate from this acoustic difference? Even if electrodes are not on temporal areas, the signal may diffuse to other regions. To remove this hypothesis, maybe it would have been useful to counterbalance the trials with a double tone, half being associated to IVM, and half to MW.
By the way, was it a silent video? Is it possible to have more details on group dances (or a sample)?
L.132-133: what the authors mean by “we continuously recorded […] for as long as possible”? Why wasn’t the whole task recorded? As I understand, the task was unlimited, but participants usually could stay vigilant for 15 minutes. Am I correct? What was the shortest and longest recording time? Was it continuous or sometimes did you have to stop the recording to make the participants vigilant (by painful stimulations for example)?
The simple motor imagery task isn’t sufficiently detailed. Was it always performed after the IVM task? If so, was there a break to be sure that subjects weren’t tired (leading to worse performances)? How long was the task? How many repetitions? Usually, the MI task is constituted of both MI and rest epochs (as the authors did in the IVM task). Was it the case here? If so, the analysis of the MI task should compare MI and rest from the same task. If there were only MI trials, I am not sure that we can compare the results with the IVM task that is rather different. In that respect, I don’t think the statistical analysis grouping IVM, MI, and MW is correct because MI comes from a different task than IVM and MW.
If I congratulate the authors for having compared their IVM task to the gold standard imagery task that is hand squeezing, I think that a condition is clearly lacking. Did the authors previously compare the condition of performing a group dance with a background video (as you did here), with the condition of performing a group dance without a background video? I think that it should be done before the comparison with MI done here, and could show the influence of the video. Finally in this study we don’t know if the impressive results come from the video or because the task of performing group dance is easier to imagine than squeezing a hand.
Other remarks:
-Signal analysis:
Why were the data segmented from [-1;6] in the MI task and [-0.5;5.5] in the IVM? Could the same time period be used because the required action is very similar?
By the way, the statistical analysis of MI seems incorrect. I think that one only need to perform the analysis on electrode C3, since subjects have to squeeze their right hand. Adding other electrodes may just add confusion and reduce the effect. Similarly, only the mu band frequency is usually interesting in this case.
Wouldn’t it be interesting to compare left and right brain regions also? In Figure 1, in MI and IVM alpha Beta, for the MCS and CG the stars are lateralized. Is there a reason?
Moreover, I don’t understand how the prediction were made. By a software with a machine learning method? By humans? If it has been done by humans, it is important to know how many humans did this and to assess the inter-experimenter reliability. At this point, I don’t understand how classification accuracy has been calculated.
-Patients criteria
L.96: Why in the exclusion criteria there is a DOC condition lasting less than three months after the brain injury? I thought that the most common criteria is less than 3 months after a non-traumatic injury, and 12 months after a traumatic injury. I formulate this remark because UWS subject 8 has the shortest disease duration. The etiology is not defined (what do T, V and A mean in table 1?), but if T means traumatic injury, then this patient had significant chances to recover compared to other participants, which may explain why he succeeded in performing the task.
-Results:
Figure1: Why alpha and beta are grouped for MI and IVM? These frequency bands have been analyzed separately? Please give the lowest and highest values of the scale.
There is a large difference in MW in the 3 groups? Couldn’t we suppose that MW condition reflects visual processing and that the 3 groups should have had the same topography?
L.186: UWS patient 8 show ERD/ERS response but it does not look really the same as MCS patients.
Minor remarks:
Title: bridging the gap. Which gap?
L.65: it is true that it is important to detect awareness to see if patients could experience pain, but not necessarily in visuomotor processes.
L.93: What SUAP mean?
L.99: What kind of visual and auditory evoked potentials?
L.103 HC subjects = control group? What is the mean age of this group? How many male and female?
Table 1: misdiagnosed patient is not highlighted in bold.
L.147: please precise frequency ranges for delta, theta, alpha, beta, gamma.
L.228: I do not agree with the authors. New HD-EEG devices are easily applicable. Moreover, one only need a very few electrodes when performing the classical MI task.
L.237: please tone down this sentence. The patient seemed to be aware according to these results.
L.242: Poor cooperation is also true with this EEG paradigm. If the patient does not want to perform the task and is always mind wandering, then he would not be classified as aware here (which is also a reason why such significance in these results is surprising).
L.270: What does the sentence “the EEG activity before the commands was unremarkable in all participants” mean?
Author Response
Dear Editor,
We want to thank you and your reviewers for the appreciation to our manuscript and the useful comments to improve its quality. We have addressed reviewers’ concerns in the present R1 response letter. Changes in the manuscript were highlighted using red characters.
To the Reviewer #2:
- This manuscript offers a new approach in the detection of covert awareness in patients with DoCs. Authors used a visuo-motor guided imagery task in 20 patients with DoCs and found that all patients with MCS generated appropriate EEG responses, as well as one patient with an unresponsive Wakefulness Syndrome (UWS). Compared to a classical motor imagery paradigm, visuo-motor guided imagery provided much better results, suggesting that this method could be used as a screening tool for patients with DoCs. First of all, I have to say that I found this article very well written. Ideas are clearly exposed and as a reader it is easy to follow the intentions of the authors. I want to congratulate the authors for this. Moreover, I found that the patients sample not so small. I know how difficult it is to recruit such patients. However, there are major concerns that stroke me throughout the manuscript and make me wonder about the validity of the results presented here.
We want to thank the reviewer for the appreciation to our manuscript and the useful comments to improve its quality.
- The results are very impressive. Despite the task being rather difficult for a patient, all patients (including MCS and UWS) had p-values below <0.1 both in the IVM task AND the MI task (table 1). I don’t know if the Bonferroni correction has any sense here since this is an exploratory study and data are analyzed at the individual level. At least, most of UWS subjects should have had p value far above 0.1 in the MI task. I don’t blame the authors for obtaining so impressive results, but necessarily one has to remove all other potential explanations that could explain why this worked so well (and also to replicate the study).
We thank the reviewer for these punctuations that significantly improved, in our opinion, the clarity of our results. We carried out the statistical analysis in the way suggested by the reviewer, i.e., without grouping IVM, MI, and MW and without Bonferroni correction, as this is an exploratory study and data are analyzed at the individual level, as correctly pointed out by the reviewer. The new statistical data seem now more in line with reviewer’s point of view, on which we agreed. Overall, our data still point out a clear difference between MCS and UWS patients in response to the diverse tasks, as well as the specific responsiveness of UWS patient no. 8.
- The fact that these results can be observed in (almost) all frequency ranges and all conditions is also very surprising.
Whole band frequency perturbation is indeed in line with our previous work on visuomotor processes in patients with DoC. As the issue of testing visuomotor processes with these types of approaches is recent, we can only make a hypothesis on the whole frequency range perturbation. This may reflect a hyper-facilitated brain responsiveness secondary to severe brain damage in the attempt to recover brain function in the framework of a large-scale connectivity breakdown that “stresses” the brain to work in many frequency ranges, as proposed by very recent data on multiplex-multilayer network analyses from our group. Lastly, such large frequency ranges entrainment may depend on the many cognitive functions that had to be entrained in order to carry out and complete the task, including sustained attention, response selection (between the tasks), language comprehension (of the task instructions), and working memory (to remember which task to do), as already outlined.
- The absence of relationship with clinical characteristics may also appear surprising.
Consistently with reviewer’s concern, we added more discussion on this issue. Beyond the already acknowledged biasing effects of drugs on cortical excitability and the similar condition of testing (non-sedated, after having received the CRS-R arousal protocol) on the lack of clinical-electrophysiological correlations in our study, it has been reported that etiology correlates with behavioral responsiveness, particularly in the long-term, in the UWS but not MCS patients. The discrepancy between our and literature data on the well-known effects of age, gender, disease duration, and initial CRS-R score on clinical-electrophysiological correlations may depend on the fact that age was rather variable, disease duration was restricted by the inclusion criteria and was higher in MCS (smaller sample) than in UWS group. As we were interested on between-group differences, some peculiar characteristics may have been lost in the analysis, which should be corroborated by further RCT.
- Of course, the main limitation I see, and that is recognized by the authors, concerns the lack of fixation and eye-tracking of patients with DoCs. In fact, from my own experience, I don’t know how possible it is to make these patients watch a video for such a long time (which does not mean it is impossible). Consequently, could the authors explain in more details what the sentence l.132-133: “Clinical inspection to ensure maintained vigilance was also conducted.” Actually mean? What did they do if the patient did not maintain vigilance?
Participants were watching a video showing group dances on a 6-inch portable screen put in front of the patient at about 15 cm. Patients’ head was held in position by a fixing device so to favor video observation. Therefore, both the distance between the screen and the face and the head-fixing device made the patients to watch the video for the time required by the experiment, also consistently with the visual scores on the subscale of the CRS-R. Furthermore, clinical inspection to ensure maintained vigilance was also conducted. In this regard, we sought for underarousal episodes, i.e., a contact between the upper and lower eyelids maintained continuously for more than 3 seconds. In such circumstance, the task was paused, the Arousal Facilitation Protocol (AFP) was administered, and then the task was resumed.
- Moreover, did the ocular artifacts were sufficiently removed after bad epoch rejection and ICA? Notably on frontal electrodes that were removed for analysis. The authors write l.145 that they obtained 70 +/- 14 epochs. Was it for IVM only or for IVM + MW? What was the percentage of rejected epochs? To finish on the question of eye movements, controls were on a chair, whereas patients were lying on a bed. Is it sure that patients (especially UWS) could efficiently watch the video? Was their head raised by some pillows for example?
We formerly found that both visual inspection and ICA were sufficient to remove ocular artifacts, notably on frontal electrodes [28-29,42]. We better detailed the numbers and percentages related to artefact removal. Finally, participants were watching a video showing group dances on a 6-inch portable screen put in front of the patient at about 15 cm. Patients’ head was held in position by a fixing device so to favor video observation. Therefore, both the distance between the screen and the face and the head-fixing device made the patients to watch the video for the time required by the experiment.
- When looking at figure 1, I am surprised that only the control group activated the occipital region (and only during MW). Shouldn’t we expect to see this region activated in all subjects while watching a movie?
We apologize for having misreported this data. We revised result description, including that there was a clear and significant activation of parietooccipital region as suggested by the parieto-occipital theta and gamma ERD. This was consistent with previous data on visuomotor stimulation in patients with DoC.
- Maybe the authors should also report the visual scores on the subscale of the CRS-R to ensure that they were able to follow a visual stimulation.
Done
- My second main concerns are related to the design of the study. How can we be sure that EEG responses are not biased by something else that covert awareness. There is a paragraph about this in the discussion but I don’t think this is enough. For example, isn’t it possible that the different EEG responses are not reflecting any perceptual process? In the IVM task they have to imagine dancing when they hear a double tone, whereas in the MW they only hear a single tone. May the signal differences originate from this acoustic difference? Even if electrodes are not on temporal areas, the signal may diffuse to other regions. To remove this hypothesis, maybe it would have been useful to counterbalance the trials with a double tone, half being associated to IVM, and half to MW. By the way, was it a silent video? Is it possible to have more details on group dances (or a sample)?
We thank the reviewer for this interesting food for thought. We took into account and discussed the concern raised by the reviewer. Furthermore, we enlarged this section with other possible biasing effects. A sample of the group dance video was also provided as supplemental video 1, as required.
- 132-133: what the authors mean by “we continuously recorded […] for as long as possible”? Why wasn’t the whole task recorded? As I understand, the task was unlimited, but participants usually could stay vigilant for 15 minutes. Am I correct? What was the shortest and longest recording time? Was it continuous or sometimes did you have to stop the recording to make the participants vigilant (by painful stimulations for example)?
We revised this misleading sentence, as correctly pointed out by the reviewer. We specified that the EEG recording lasted 16±3 minutes (range 13-22 minutes). All patients were able to carry out the task for this time span. The duration of recording depended on the frequency of underarousal, which also determined the amount of delivered AFP. The task was conducted until the underarousal episodes began to occur continuously.
- The simple motor imagery task isn’t sufficiently detailed. Was it always performed after the IVM task? If so, was there a break to be sure that subjects weren’t tired (leading to worse performances)? How long was the task? How many repetitions? Usually, the MI task is constituted of both MI and rest epochs (as the authors did in the IVM task). Was it the case here? If so, the analysis of the MI task should compare MI and rest from the same task. If there were only MI trials, I am not sure that we can compare the results with the IVM task that is rather different. In that respect, I don’t think the statistical analysis grouping IVM, MI, and MW is correct because MI comes from a different task than IVM and MW.
Accordingly, we better detailed the MI procedure, which however reflected that of IVM-MW. In particular, we already stated that two types of motor imagery task were conducted in the same day of recording with a 60-minute period of rest. The order of the two experimental sessions was random but globally counterbalanced. As correctly suggested, we carried out the statistical analysis in the way suggested by the reviewer, i.e., without grouping IVM, MI, and MW.
- If I congratulate the authors for having compared their IVM task to the gold standard imagery task that is hand squeezing, I think that a condition is clearly lacking. Did the authors previously compare the condition of performing a group dance with a background video (as you did here), with the condition of performing a group dance without a background video? I think that it should be done before the comparison with MI done here, and could show the influence of the video. Finally in this study we don’t know if the impressive results come from the video or because the task of performing group dance is easier to imagine than squeezing a hand.
We thank the reviewer for this interesting hint. We performed the suggested experiment in a subgroup of the patients and the CG individuals (we could not study the entire sample because some participants were not yet examinable, also due to SARS-CoV-2 pandemic). We found that the advanced-MI task yielded EEG responses consistent with motor imagery only in a portion of CG individuals, in a few patients with MCS, and in none of those with UWS. Conversely, all HC and MCS participants showed EEG responses to IVM task consistent with motor imagery processes. These responses were even milder and less distributed concerning simple-MI task. Therefore, our result indicate that EEG responses came from the video provision and not because the task of performing group dance was easier to imagine than squeezing a hand, as correctly hypothesized by the reviewer.
- Other remarks:
- -Signal analysis:
- Why were the data segmented from [-1;6] in the MI task and [-0.5;5.5] in the IVM? Could the same time period be used because the required action is very similar?
We corrected the misreported data.
- By the way, the statistical analysis of MI seems incorrect. I think that one only need to perform the analysis on electrode C3, since subjects have to squeeze their right hand. Adding other electrodes may just add confusion and reduce the effect. Similarly, only the mu band frequency is usually interesting in this case.
The analysis was rebuilt consistently with reviewer’s suggestions.
- Wouldn’t it be interesting to compare left and right brain regions also? In Figure 1, in MI and IVM alpha Beta, for the MCS and CG the stars are lateralized. Is there a reason?
Accordingly, data analysis was rebuilt by comparing left and right brain regions. EEG changes during simple-MI task were focused on left central electrodes, as foreseeable. The same occurred for alpha and beta frequency ranges during the IVM. This different result as compared to the literature data may depend on several issues, including the different types of motor imagery, extent of movement duration, feedback interference of muscle network activation during motor imagery task, activation of subcortical structures (that are critically damaged in DoC, especially post-anoxic), and motor-cognitive disability degree. Furthermore, it is likely that the MNS largely contributed to generate these EEG responses. Even though the changes in corticomotor excitability during action observation (alpha/beta ERS/ERD) is the established and accepted feature of MNS activation, and the timing and lateralization of the EEG responses expressing the potential contribution of MNS to motor imagery is yet obscure, the plasticity of the MNS (theta/gamma ERS/ERD) could have resulted in a lastingly and lateralized increased corticomotor excitability that was greater during IVM than during MI.
- Moreover, I don’t understand how the prediction were made. By a software with a machine learning method? By humans? If it has been done by humans, it is important to know how many humans did this and to assess the inter-experimenter reliability. At this point, I don’t understand how classification accuracy has been calculated.
We added more details on this issue. We specified that the accuracy of the ERS/D in predicting group membership was estimated as the number of correct predictions (each ERS/ERD) from all predictions made (all EEG trials). In this regard, the logarithmic band power for each frequency band was calculated in consecutive, non-overlapping 1-sec sample-by-sample in each trial (by band-pass filtering, squaring and averaging the samples) compared to a reference (i.e., 1 s before the command delivery). A single value was sampled at the middle of each time-period and was thus used in the classification accuracy analysis. To this end, we used a linear discriminant analysis classifier, with a nested block-wise cross-validation (10×10 inner fold; leave-one-out-block outer fold) being applied (to avoid overfitting). The experimenters who analyzed the EEG data were blind to patients’ behavioral responsiveness level.
- -Patients criteria
- 96: Why in the exclusion criteria there is a DOC condition lasting less than three months after the brain injury? I thought that the most common criteria is less than 3 months after a non-traumatic injury, and 12 months after a traumatic injury. I formulate this remark because UWS subject 8 has the shortest disease duration. The etiology is not defined (what do T, V and A mean in table 1?), but if T means traumatic injury, then this patient had significant chances to recover compared to other participants, which may explain why he succeeded in performing the task.
We agree on reviewer’s concern concerning DOC condition duration. However, we applied this inclusion criterion to be consistent with all our previous work concerning visuomotor integration. Actually, this function recovers usually within the first three months, so we excluded this time span to ensure to work on “chronic” DoC, at least concerning visuomotor function (added references). It is also true that subject 8 has the shortest disease duration but there were other two patients in UWS with a similar disease duration and etiology who did not show any positive response to the VM tasks. However, further larger sample studies may clarify this interesting issue.
- -Results:
- Figure1: Why alpha and beta are grouped for MI and IVM? These frequency bands have been analyzed separately? Please give the lowest and highest values of the scale.
Fig. 1 was rebuilt consistently with the new data analysis suggested by the reviewer.
- There is a large difference in MW in the 3 groups? Couldn’t we suppose that MW condition reflects visual processing and that the 3 groups should have had the same topography?
Each MW condition was compared with its respective active condition (IVM, simple-MI, and advanced-MI) consistently with the new analysis suggested by the reviewer. When looking at between-group difference in MW (not a primary analysis) a significant difference emerged, which is likely to depend on the fact that the participant was not instructed on what to do during the MW condition except to not image group dancing, thus reflecting a deteriorated resting state network activity (as suggested by preserved power distribution in CG sample).
- 186: UWS patient 8 show ERD/ERS response but it does not look really the same as MCS patients.
Accordingly, we toned down the sentence.
- Minor remarks:
- Title: bridging the gap. Which gap?
We revised the title avoiding misleading words.
- 65: it is true that it is important to detect awareness to see if patients could experience pain, but not necessarily in visuomotor processes.
Checked and corrected in “visuomotor as well as other sensorimotor processes”
- 93: What SUAP mean?
Checked and corrected in “long-term rehabilitation unit” (SUAP is an Italian acronym).
- 99: What kind of visual and auditory evoked potentials?
conventional visual and auditory evoked potentials
- 103 HC subjects = control group? What is the mean age of this group? How many male and female?
We added the missing information
- Table 1: misdiagnosed patient is not highlighted in bold.
Checked and corrected
- 147: please precise frequency ranges for delta, theta, alpha, beta, gamma.
Done
- 228: I do not agree with the authors. New HD-EEG devices are easily applicable. Moreover, one only need a very few electrodes when performing the classical MI task.
Accordingly, the sentence was removed.
- 237: please tone down this sentence. The patient seemed to be aware according to these results.
The sentence was rewritten as suggested.
- 242: Poor cooperation is also true with this EEG paradigm. If the patient does not want to perform the task and is always mind wandering, then he would not be classified as aware here (which is also a reason why such significance in these results is surprising).
We agree with reviewer’s concern. This issue was discussed in the limitation section
- 270: What does the sentence “the EEG activity before the commands was unremarkable in all participants” mean?
The sentence was revised since misleading in “…specific EEG activities before command delivering were not appreciable…”
Kindest regards,
the authors
Round 2
Reviewer 2 Report
The authors did a great work in a limited time. I still have concerns about the design of the experiment, but that could not be changed easily. As a consequence, I thaink that the manuscript can be accepted in its present form.